# Effect of Ni on the Suppression of Sn Whisker Formation in Sn-0.7Cu Solder Joint

**DOI:** 10.3390/ma14040738

**Published:** 2021-02-05

**Authors:** Aimi Noorliyana Hashim, Mohd Arif Anuar Mohd Salleh, Andrei Victor Sandu, Muhammad Mahyiddin Ramli, Khor Chu Yee, Noor Zaimah Mohd Mokhtar, Jitrin Chaiprapa

**Affiliations:** 1Centre of Excellence Geopolymer and Green Technology, Universiti Malaysia Perlis (UniMAP), Taman Muhibah, Jejawi, Arau, Perlis 02600, Malaysia; aimiliyana@unimap.edu.my (A.N.H.); noorzaimah3287@gmail.com (N.Z.M.M.); 2Faculty of Chemical Engineering Technology, Universiti Malaysia Perlis (UniMAP), Taman Muhibah, Jejawi, Arau, Perlis 02600, Malaysia; 3Faculty of Materials Science and Engineering, Gheorghe Asachi Technical University, D. Mangeron 41, 700050 Iasi, Romania; 4Romanian Inventors Forum, Str. Sf. P. Movila 3, L11, 700089 Iasi, Romania; 5School of Microelectronic Engineering, Pauh Putra Campus, University Malaysia Perlis (UniMAP), Arau, Perlis 02600, Malaysia; mmahyiddin@unimap.edu.my; 6Faculty of Engineering Technology (FETech), Universiti Malaysia Perlis (UniMAP), Level 1, Block S2, UniCITI Alam Campus, Sungai Chuchuh, Padang Besar, Perlis 02100, Malaysia; cykhor@unimap.edu.my; 7Synchrotron Light Research Institute (SLRI), 111 University Avenue, Muang District, Nakhon Ratchasima 30000, Thailand; jitrin@slri.or.th

**Keywords:** Sn whisker, reliability, lead-free solder, Sn-0.7Cu-0.05Ni, whisker growth, mitigation, interconnects, soldering

## Abstract

The evolution of internal compressive stress from the intermetallic compound (IMC) Cu_6_Sn_5_ growth is commonly acknowledged as the key inducement initiating the nucleation and growth of tin (Sn) whisker. This study investigates the effect of Sn-0.7Cu-0.05Ni on the nucleation and growth of Sn whisker under continuous mechanical stress induced. The Sn-0.7Cu-0.05Ni solder joint has a noticeable effect of suppression by diminishing the susceptibility of nucleation and growth of Sn whisker. By using a synchrotron micro X-ray fluorescence (µ-XRF) spectroscopy, it was found that a small amount of Ni alters the microstructure of Cu_6_Sn_5_ to form a (Cu,Ni)_6_Sn_5_ intermetallic layer. The morphology structure of the (Cu,Ni)_6_Sn_5_ interfacial intermetallic layer and Sn whisker growth were investigated by scanning electron microscope (SEM) in secondary and backscattered electron imaging mode, which showed that there is a strong correlation between the formation of Sn whisker and the composition of solder alloy. The thickness of the (Cu,Ni)_6_Sn_5_ IMC interfacial layer was relatively thinner and more refined, with a continuous fine scallop-shaped IMC interfacial layer, and consequently enhanced a greater incubation period for the nucleation and growth of the Sn whisker. These verification outcomes proposes a scientifically foundation to mitigate Sn whisker growth in lead-free solder joint.

## 1. Introduction

Presently, the lead (Pb)-free solder has become a top priority in electronic industries. This is concurrent with the enactment of the EU Directive 2002/95/EC that states the use of lead (Pb) must either be eliminated or limited in a very low concentration in electrical and electronic equipment (EEE) [1,2,3,4,5,6,7,8]. The regulation to limit the use of lead in solder has brought back the Sn whiskers phenomenon and triggered numerous system failures [1,3,8,9,10,11,12,13]. Furthermore, the trends of electronics miniaturization in microelectronics packages have resulted in a serious reliability issue of lead-free solder in electronic devices [6,8,14,15]. Numerous studies have already been carried out, and researchers agree that the main inducement factor for Sn whisker nucleation and growth due to the stress progression from the development of a SnCu intermetallic compound (IMC) layer at room temperature [2,3,8,11,13,16,17,18,19]. A Cu_6_Sn_5_ IMC interfacial layer begins to form at the solder/substrate interface as the copper (Cu) atoms diffuse into the tin (Sn) layer. The growth of intermetallic particles into the tin (Sn) layer to reach the solution saturation limit can lead to a significant effect on thermal expansion mismatch, and thus create a residual stress gradient at the solder/substrate interface [20]. 

Illes et al. [21] surmise that the main source for whiskers growth is the non-uniform shape of IMC interfacial layers and expansion of the IMC interfacial layer at the solder/substrate interface layers [8,21,22]. As explained by Kim et al. [22], the Sn whisker prone to grow with irregularly shaped due to high residual compressive stresses concentrated at IMC interfacial layer. The natural surroundings of the Sn whisker development are spontaneous and slow-moving, and consequently, an assessment of the Sn whisker kinetic growth can often result in an unreliable and unpredictable growth rate [8]. It is most beneficial to have access to a fundamental study of the Sn whiskers phenomenon in a well-ordered stimulus condition [8]. Multiple studies have been carried out to study the acceleration behavior of Sn whisker formation and growth through mechanical stress induced [13,23], but only few works have shown continuous mechanical stress induced to accelerate Sn whisker nucleation and growth [2,8]. From previous studies, both nucleation and growth of Sn whisker are capable to form, even though there is no external induced stress applied. However, as determined by Lin et al. [2], the Sn whisker growth is longer and denser, with continuous mechanical stress induced [2]. In addition, Jagtap et al. [23] also claimed that the stress gradient in the solder/substrate interface layers accelerates the nucleation and growth of Sn whiskers with time evolvements under continuous mechanical stress induced by indentation test fixture [23].

Many existing studies have verified that lead (Pb) noticeably has a great suppression significance on the Sn whisker nucleation and formation by enhancing the stress relaxation behavior in the solder layer [3,9,18,20]. Much effort has been aimed at reliability improvement techniques, such as adding a third element [10,24,25], but a spot-on lead-free solder replacement has not yet been found to suppress whiskers formation as effectively as the addition of Pb in solder [14,22,26]. Many studies found similar results that specify the mechanism for the stress relaxation enhancement in solder layers with the addition of micro-alloying elements in the solder joint [11,15]. Considering the effects of adding micro-alloying elements in lead-free solder can certainly lead to the formation of new compounds in the IMC interfacial layer in addition to Cu_6_Sn_5_, and can significantly modify the properties of the IMC interfacial layer [7,27].

Micro-alloying with Ni indicates noticeably high solubility in Cu_6_Sn_5_ and pointedly influences the microstructure of the lead-free solder joint. Sn-0.7Cu-0.05Ni enables the nucleation of Cu_6_Sn_5_ and is possible to solidify as a eutectic, similar to tin lead solder [28]. It has been reported that Ni is able to reduce both magnitude [29] and expansion discontinuity of thermal expansion of Cu_6_Sn_5_ with the polymorphic phase transformation, mainly along the a-axis [27]. The distinctive properties of Sn-0.7Cu-0.05Ni could diminish the internal stress residual in solder joints, particularly during the thermal cyclic condition [27]. It has also been discovered that Ni micro-alloying can generate stabilization at high-temperature (around 250 °C) phase transition of the (Cu,Ni)_6_Sn_5_ IMC hexagonal phase [30] and is able to increase the stress relaxation in the layer with the formation of fine, needle-like (Cu,Ni)_6_Sn_5_ at the Sn–Cu interface [8]. In addition, Sn-0.7Cu-0.05Ni has good fluidity and oxidation resistance condition compared to the Sn–0.7Cu alloy, and also, (Cu,Ni)_6_Sn_5_ grows as an initial phase during solidification [27]. Therefore, it is remarkable to determine the correlation between the (Cu,Ni)_6_Sn_5_ IMC interfacial layer and Sn whisker formation with regard to the IMC interfacial layer suppression and stress relaxation enhancement for both the nucleation and growth of the Sn whisker.

This study investigated the effects of Sn-0.7Cu-0.05Ni solder joints on the nucleation and growth of the Sn whisker under continuous mechanical stress induced using scanning electron microscope (SEM), synchrotron microbeam X-ray fluorescence (µ-XRF) spectroscopy and atomic force microscopy (AFM). The findings resulted an in-depth understanding of the suppression effects of the (Cu,Ni)_6_Sn_5_ IMC interfacial layer on the nucleation and growth of the Sn whisker in the Sn-0.7Cu-0.05Ni solder joint.

## 2. Experimental Procedures 

### 2.1. Materials

The raw materials of Sn-0.7Cu and Sn-0.7Cu-0.05 Ni solder alloy were provided by Nihon Superior (M) Sdn. Bhd., Ipoh, Perak, Malaysia. The solder alloys were prepared by melting ingots in a solder pot at a temperature of 265 °C. The industrial high-purity Cu above 99.9% were used as substrates with dimensions of 1.5 × 1.5 cm^2^ and 1.0 cm in thickness. Table 1 depicts the elemental composition analysis by arc spark spectrometer used throughout the study.

### 2.2. Sample Preparation

The hot-dipping approach was used to prepare Sn-0.7Cu and Sn-0.7Cu-0.05Ni solder joints. The Cu substrate was manually dipped into acid liquid solution comprised of 5% hydrochloric acid (HCl) and deionized water at ambient temperature for 5 min to get rid of surface oxides film and then rinsed with acetone (C_3_H_6_O) and distilled water, and subsequently air-dried to remove oil contamination. To enhance the soldering and for the removal of oxidation, substrate was immersed in a flux solution. A standard B-type solution flux based on Japanese Industrial Standards (JIS Z3198-4) was applied which contains of mixtures of resin, isopropyl alcohol and diethylamine hydrochloride.

The Cu substrate was then positioned on the pneumatic angular gripper of an automated solder-dip machine and then dipped in the molten solder bath with an immersion dwell time of 2 s and speeds of withdrawal of 10 mm/s. The temperature of the molten solder bath was held at 265 °C. During withdrawal, the surface of the solder coating was blown off using a compressed air knives blower with controllable angle and speed of pressure to remove the excess solder coating and to improve smoothness of the solder coating layer. Then, the samples were air-cooled and water-rinsed using an ultrasonic bath. Around 10–20 µm thickness of solder coating layer was applied on the substrate. Figure 1 shows the schematic diagram of the solder hot-dipping process. The solder hot-dipping temperature profile process is presented in Table 2. After the hot-dip coating, samples were aged at room temperature for 48 h to saturate the internal residue stress induced by IMC formation.

### 2.3. Testing and Characterization Method

The micro-indentation test was employed to stimulate the Sn whiskers nucleation and growth through continuous mechanical stress induced with a constant load. This accelerated testing method increases the distribution of compressive stress that is transmitted to the weaker grain boundaries. Subsequently, the stress could be accumulated and promotes the nucleation and growth of Sn whiskers from surface flaws, such as from voids and cracks. Figure 2 illustrates the schematic of micro-indentation apparatus used in this study. The test apparatus was located in the clean room at ambient temperature. Periodic observations of the samples under continuous mechanically induced stress on the nucleation and growth of Sn whiskers were employed for each sample throughout 20 days using a JEOL JSM 6460LA scanning electron microscope (SEM), JEOL Ltd. Tokyo, Japan in secondary and backscattered electron imaging mode at an accelerating voltage at 20 KV. The Sn whisker morphology was observed with a 30° tilt angle. This investigation outcomes presented as a parameters to assess the trends of the Sn whisker nucleation and growth in Sn-0.7Cu-0.05Ni.

The length and density of the Sn whisker distribution were considered based on Whisker Standards (JESD22-A121A) of the Joint Electron Device Engineering Council (JEDEC) [8,15] by picking five equivalent sampling areas for each sample, using a Java image processing program, image-J Software with an average value of five interpretations per analysis. The measurement of axial Sn whisker length is calculated from the bottom of the Sn whisker growth to the whisker tip, whereas for bend, the Sn whisker is composed of the straight section of the whisker, as shown in Figure 3. The precision for the Sn whisker length and Sn whisker density calculation was about ±5%. It is essentially reliant on the magnification ratio of the particular image micrograph. The density of the Sn whisker is defined as the total number of Sn whiskers’ growth per unit area of the observed area and considered in a unit of pcs/µm^2^. The error bars were measured from the standard deviation (SD) of the Sn whiskers’ distribution. 

The samples were then mounted using a mixture of resin and hardener to cross-section, followed by a grinding and polishing process in order to measure an average IMC interfacial thickness of the solder joint. The IMC interfacial layer thickness (T), is equal to total IMC area (A), divided by total IMC length (L), and the unit is expressed in µm. With the purpose of observing the top view of the three-dimensional (3D) morphology of the IMC interfacial layer, the solder joint was then entirely deep-etched, consuming a mixed solution of 2% ortho-nitrophenol, 5% sodium hydroxide and 93% of distilled water at a temperature 70 °C to selectively eliminate the tin (Sn) element of the solder alloys. The shape, thickness and root mean square (RMS) roughness of the IMC interfacial layer were analyzed by JEOL JSM 6460LA scanning electron microscope (SEM) by JEOL Ltd. Tokyo, Japan, high-resolution of atomic force microscope (AFM) by Bruker Corporation, Billerica, MA, Statele Unite and Profilm online web based software program by Kla corporation, Milpitas, CA, Statele Unite. The procedure of AFM comprises scanning a sharp tip incorporated in a flexible cantilever across the sample surface, monitoring the tip–sample interaction to produce a 3D topographic micrograph figure of the surface. This technique permits to compute the surface properties, such as RMS roughness, particle size, step height, etc. Surface roughness provides the quantitative data of the surface features based on the statistical deviation from average height. Besides, AFM data able to generate a data containing information of surface roughness in a graphic arrangement to show the growth features of the IMC interfacial layer.

A methodology observation through synchrotron microbeam X-ray fluorescence (µ-XRF) spectroscopy was accomplished for elemental mapping to detect small trace metal element distribution, such as Cu (70 ppm) and Ni (500 ppm), in solder alloys. This analysis was performed at the Synchrotron Light Research Institute (SLRI), Kanchanaburi, Thailand. The BL6b beamline exploited the continuous synchrotron radiation that is emitted from the bending magnet of the 1.2 GeV storage ring with a micro-focused beam of 30 × 30 µm^2^. Imaging analysis was processed using the PyMca 5.6.3-win64.exe software. In order to observe the elemental distribution mapping of the lead-free solder joint, samples of micro-XRF were mounted using a mixture of resin and hardener and polished in the cross-sectioned condition. The thicknesses of all solder coating layers were synthesized in the range from 200 to 300 µm due to the limits of spatial resolution level, with X-ray beam spot size in the order of 50 µm and above.

## 3. Results and Discussion

The surface morphologies of Sn-0.7Cu and Sn-0.7Cu-0.05Ni solder joints were periodically observed from day 1 up to day 20 under continuous mechanically induced stress. The observation of Sn whisker growth behavior resulted from two types of solder joint, Sn-0.7Cu and Sn-0.7Cu-0.05Ni, as shown in Figure 4. The observed distribution of Sn whisker growth was increasing over storage time. In Figure 4a,b, hillock- or nodule-shaped Sn whiskers were noticed on both of the solder joints on day 1. The Sn whiskers appear to progressively extrude from the surface of Sn-0.7Cu and Sn-0.7Cu-0.05Ni. It was evidently seen that the initial process of whiskering is a nucleation that protruded as a hillock or nodule shape on the surface. In response to the stress gradient of the IMC interfacial layer, hillock could probably to grow as a risky long filament-shaped Sn whisker [31]. With an increase of storage time from day 5 to day 20 (Figure 4c–f), the hillock-shaped Sn whisker rapidly grew and became larger on the surface of the solder joint. In contrast, Sn whiskers’ growth were found greater on the Sn-0.7Cu solder with the growth of filament-shaped and bent-shaped Sn whiskers, while on the surface of the Sn-0.7Cu-0.05Ni solder joint, the hillock shape was observed.

It can be observed that the growth trend of Sn whiskers’ length and density on Sn-0.7Cu and Sn-0.7Cu-0.05Ni solder joints increased as the storage time increased. As shown in Figure 5, the Sn whiskers’ length on Sn-0.7Cu was observed to be higher than on the Sn-0.7Cu-0.05Ni solder joint. The size of the average Sn whisker length on Sn-0.7Cu increased more rapidly at day 3 until day 20. This occurrence tends to contribute a high tendency to short-circuit fine pitch components in electronic packages and is considerably dangerous to microelectronic reliability.

It can also be noticed that Sn whisker tendency, with respect to Sn whisker density, significantly increased with storage time and finally reached a saturated condition at day 20 for both Sn-0.7Cu and Sn-0.7Cu-0.05Ni solder joints. From this investigation, as presented in Figure 6, the average density of Sn-0.7Cu was obviously higher than the Sn-0.7Cu-0.05Ni solder joint, where the Sn-0.7Cu-0.05Ni solder joint has been found to be less prone to Sn whiskers. This observation also indicates that the density of Sn whiskers was greater as storage time progressed. The trend of Sn whisker nucleation and growth corresponds to the IMC interfacial growth and residual stress evolution that increases over time to reach the solution saturation [32]. Therefore, it could be concluded that there is a significant relationship between the Sn whisker growth and composition of the lead-free solder joint. The Sn-0.7Cu-0.05Ni solder joint has an obvious suppression effect by diminishing the susceptibility for Sn whisker growth and increase the incubation time for nucleation of Sn whiskers.

Since the residual stress and IMC interfacial reaction cannot be regulated, a methodical observation of Sn whisker nucleation and growth is very challenging. Apparently, the propensity for Sn whisker nucleation and growth is increased with the implementation of both internal residual stress and mechanical compressive stress induced in the solder layer. To generate well-regulated inducement of Sn whisker growth, a continuous mechanical compressive stress was induced in all samples of the Sn–Cu solder joint to accelerate the Sn whiskers’ growth by micro-indentation. Figure 7 show SEM images with a 30° tilt angle of the surface of the Sn-0.7Cu-0.05Ni solder joint that have been indented for 12 h. The indentation area indicates a localized effect and has a diameter of approximately 83.94 µm. It was observed that the hillock- or nodule-shaped Sn whisker growth was most abundant nearby the indentation area. The length of the hillock shape was found to vary from 0.5 to 3 µm in the Sn-0.7Cu-0.05Ni solder joint.

Regarding the morphology of Sn whisker nucleation growth, the primary nucleation takes place with the growth of a hillock or nodule shape, followed by protrusion of Sn whiskers. The nucleation kinetics of the Sn whisker revealed here are consistent with previous work [23] under a constant continuous mechanical stress using a clamping fixture, wherein the compressive stress extends into the region away from the punch, organized by the relaxation processes in the solder layer with the growth of the Sn whisker. The results revealed that the initiation of Sn whiskers’ growth is also consistent with Fei et al.’s [19] study. They claimed that a primary stage in the Sn whisker formation is nucleation, which is a Sn whisker extruding position. In addition, according to Skwarek et al. [33], the Sn whiskers grew out from the whole grains of the solder/substrate interface, not from the grain boundaries.

The composition and distribution analysis of Ni, Cu and Sn alloying elements in the Sn-0.7Cu-0.05Ni solder joint using synchrotron micro-XRF mapping are presented in Figure 8. The micro-XRF mapping area is ~450 × 1750 µm. The mapping of the blue area indicates low concentration or no element in the solder joint. The mapping of green and yellow areas indicates intermediate concentration, and the mapping of the red area indicates high concentration of specific composition. The small, well-distributed Ni and Cu have been found at primary IMC of the Sn-0.7Cu-0.05Ni solder joint. It is direct confirmation from the mapping result that a small amount of Ni (500 ppm) alters the microstructure of Cu_6_Sn_5_ to form a (Cu, Ni)_6_Sn_5_ intermetallic layer. The formation of new compounds in IMC interfacial layers is very significant to the phase equilibria of the system, and in addition, is able to modify the properties of the IMC interfacial layer. In the presence of Ni in Sn-0.7Cu, the Cu_6_Sn_5_ interfacial IMC layer takes the form of the (Cu, Ni)_6_Sn_5_ interfacial IMC layer, where Ni atoms occupy copper lattice sites [34]. A previous study documented that Ni is able to enhance the IMC interfacial formation with a fine scallop shape during liquid–solid interfacial reactions [27].

According to the literature [35], Ni is dispersed in a relatively homogeneous manner through the Cu_6_Sn_5_ IMC interfacial layer. However, in this study, Ni distribution was not fitting at the IMC interfacial layer due to the limits of spatial resolution level, with X-ray beam spot size at the order of 50 µm and above. It is believed that homogeneous concentration of Ni in Cu_6_Sn_5_ IMC Ni sustains the phase stability in the hexagonal form of the (Cu, Ni)_6_Sn_5_ IMC interfacial layer and prevents the transformation of polymorphic phase (hexagonal to monoclinic) of Cu_6_Sn_5_ [35]. This phase stability will minimize the stress concentrations in the IMC layer, and subsequently reduce the Sn whiskering potential on the surface of the solder joint.

The morphology of Sn whiskers was intensely inclined to the shape and thickness of the interfacial intermetallic layer (IMC) [3,25]. A quantitative analysis was done to examine the correlation of the Sn whiskers’ formation and the IMC interfacial layer using SEM in secondary and backscattered electron imaging mode at an accelerating voltage of 20 kV. An image structure of the IMC interfacial layer that formed in Sn-0.7Cu and Sn-0.7Cu-0.05Ni solder joints is presented in Figure 9. It can be observed that from the cross-sectioned view of the morphology, the Cu_6_Sn_5_ IMC interfacial layer on Sn-0.7Cu (Figure 9a) is rougher than the (Cu, Ni)_6_Sn_5_ IMC layer on Sn-0.7Cu-0.05Ni (Figure 9b). The (Cu, Ni)_6_Sn_5_ IMC interfacial layer results in a finer grained, more continuous layer with stable shape of the IMC interfacial layer. Moreover, regarding the thickness of the IMC interfacial layer, the 1.76 µm thick scalloped Cu_6_Sn_5_ IMC layer was higher compared to the 1.28 µm thick (Cu, Ni)_6_Sn_5_ IMC layer, whereas Ni micro-alloying changed the thickness of the IMC interfacial layers significantly and revealed an approximately uniform height of the IMC grains.

Sn-0.7Cu-0.05Ni has a significant implication on the Sn whisker formation and growth behavior. The outcomes of this work appear to validate that the Sn whisker susceptibility is associated with the solder composition (refer to Figure 4) as well as the IMC interface topography. From the top view in Figure 9c and d, it can be noticeably observed that there is strong correlation between morphology and size of IMC interfacial intermetallic grains and compositions of the solder alloys. At the interface of solder/substrate in the Sn-0.7Cu-0.05Ni solder joint, significantly fine and almost uniform IMC grains formed compared to the interface of the solder/substrate in the Sn-0.7Cu solder joint. Remarkably, the grain size of (Cu,Ni)_6_Sn_5_ is significantly smaller than the Cu_6_Sn_5_ IMC interfacial layer. This means that the Ni micro-alloying decreased the IMC grain size. The uniform and smaller average size of IMC grains may possibly result in a lower residual stress and suppress the propensity of Sn whisker formation [33].

The significance of the IMC interfacial layer growth directly creates internal stress in the grain boundary of the solder layer, which can induce Sn whisker growth, as illustrated schematically in Figure 10. The rapid dissolution and growth of the thick scallop shape of the IMC interfacial layer with pyramid-like IMC grains significantly produced higher cumulative stress in the IMC interfacial layer and can lead to the formation of Sn whiskers [3]. In contrast, the thinner scallop shape of IMC interfacial layers can lead to lower stress of the IMC interfacial layer and thus suppress the growth of Sn whiskers [4]. This is also consistent with the work of Chason et al. [11], who showed that the stress relaxation processes are significantly more rapid and greater for thicker Sn solder. Therefore, the morphology structure of the (CuNi)_6_Sn_5_ IMC interfacial layer was relatively thinner and more refined, with a fine scallop shape thus suppressed the thickness of the IMC interfacial layer and enhanced longer incubation period for the formation and growth of Sn whiskers. Consistent with Illes et al. [21], the mitigation of Sn whisker formation and growth can be achieved by reducing the Sn whisker incubation time, which produces thicker layers of lead-free solder, since this formation improves the stress relaxation properties.

Figure 11 indicates the topographies of IMC interfacial properties of deep-etched Sn-0.7Cu and Sn-0.7Cu-0.05Ni solder joints by AFM. The scale of contrast of Cu_6_Sn_5_ and (Cu,Ni)_6_Sn_5_ IMC interfacial layers is 2.5 µm and 2.0 µm. The higher contrast enhanced the number of thicker and higher peaks of IMC grain that formed. It was observed that the three-dimensional shape morphology of the scallop-shaped Cu_6_Sn_5_ IMC interfacial layer on Sn-0.7Cu (Figure 10a) was significantly thicker and coarser compared to the (Cu,Ni)_6_Sn_5_ IMC interfacial layer on Sn-0.7Cu-0.05Ni (Figure 10b). These results corresponded with a quantitative analysis of the SEM, as shown in Figure 9. Comparing the 0.05 wt.% Ni addition to Sn-0.7Cu, the formation of the (Cu,Ni)_6_Sn_5_ IMC layer was relatively thinned and had a finer scallop shape, which has been found to be less prone to whiskers.

Investigation of the Cu_6_Sn_5_ and (Cu,Ni)_6_Sn_5_ IMC interfacial layer was further carried out by measuring the surface roughness values and the highest peak of the IMC interfacial layer. Surface roughness is an observation parameter for describing the surface features and is commonly signified in terms of the statistical deviation from average height [8]. It was established that the surface roughness values of the IMC interfacial layers Cu_6_Sn_5_ and (Cu,Ni)_6_Sn_5_ were 1.83 and 1.26 µm, respectively. Moreover, it was observed that the highest peak value of the Sn-0.7Cu IMC interfacial layer was ~2.641 μm, whereas the highest peak value of the Sn-0.7Cu-0.05Ni IMC interfacial layer was lower, at ~2.126 μm. This implies that a more uneven, higher peak of the IMC interfacial layer was observed to be more susceptible to Sn whisker nucleation and growth. This outcome indicates that Ni micro-alloying plays an important role in the IMC interfacial layer. The thinner thickness and fine scallop-shaped formation of the (Cu,Ni)_6_Sn_5_ IMC layer enhances stress relaxation and suppress the nucleation and growth of the Sn whisker.

Similar results were found in studies by Chason et al. [11], that validate that the stress relaxation is suppressed with small grain size of IMC interfacial layers. The growth of the IMC interfacial layer can have a significant impact on the residual stress of the solder joint as the stress gradient in the solder joints is induced by volume expansion of polymorphic phase transformation. It has been stated that (Cu,Ni)_6_Sn_5_ greatly reduced the volume change associated with the polymorphic phase transformation of Cu_6_Sn_5_ [27]. This is also consistent with a finding by Nogita et al. [29], who suggest that the phase stabilization of the hexagonal (Cu,Ni)_6_Sn_5_ may possibly prevent volume changes of the IMC interfacial layer that could provide the stress relaxation in the solder layer [8,36].

## 4. Conclusions

The effect of the Sn-0.7Cu-0.05Ni solder joint on the nucleation and growth of Sn whiskers under continuous mechanically induced stress was investigated. The correlation of the Sn whiskers’ growth and the IMC interfacial layer was also established. The following conclusions can be drawn:There was a noticeable correlation between the Sn whisker growth and the composition of the lead-free solder alloy. The Sn-0.7Cu-0.05Ni solder joint had a great suppression effect on the nucleation and growth of Sn whiskers of the Pb-free solder joint.The small amount of Ni addition (~500 ppm) was able to alter the microstructure of Cu_6_Sn_5_ to form a (Cu,Ni)_6_Sn_5_ IMC intermetallic layer, and it is very significant to the nucleation and growth of Sn whiskers.The methodic structure of the (Cu,Ni)_6_Sn_5_ IMC interfacial layer was relatively thinner and more refined, with a continuous fine scallop-shaped IMC interfacial layer consequently enhanced a greater incubation period for the nucleation and growth of the Sn whisker.

## Figures and Tables

**Figure 1 materials-14-00738-f001:**
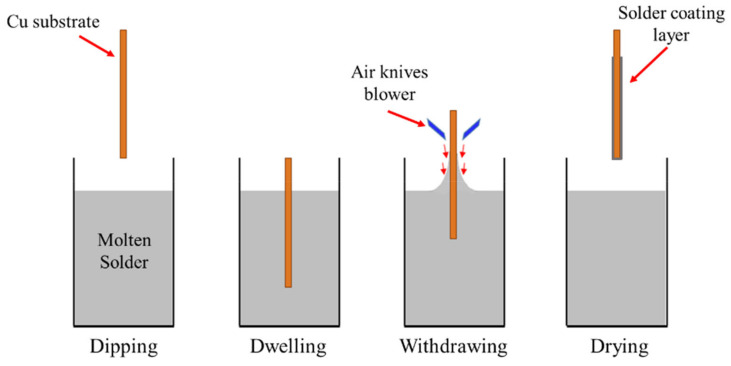
Schematic diagram of the hot-dipping process.

**Figure 2 materials-14-00738-f002:**
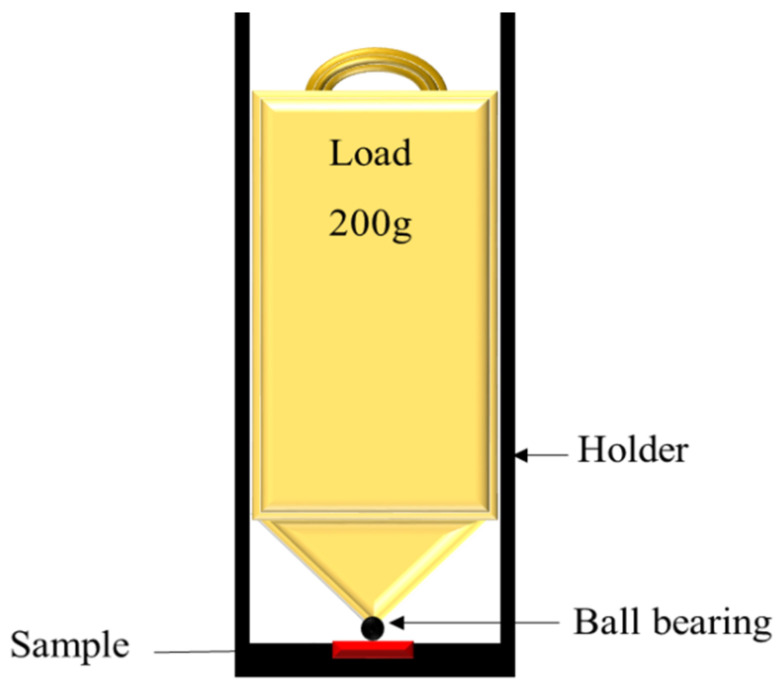
Schematic diagram of micro-indentation test apparatus.

**Figure 3 materials-14-00738-f003:**
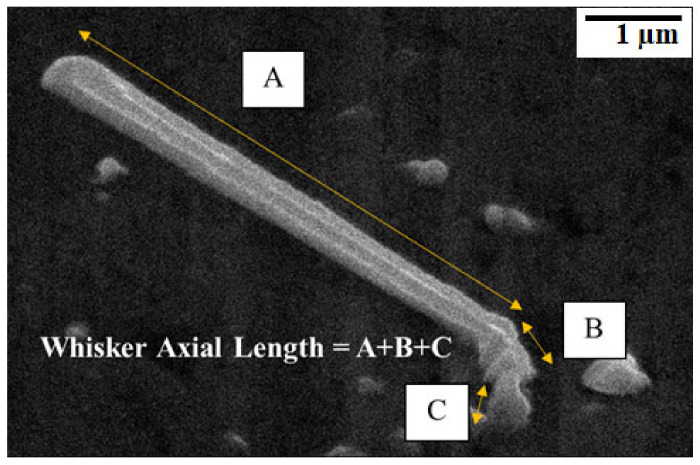
A method to calculate total axial Sn whisker length by adding all straight section of Sn whisker based on the JEDEC whisker standards.

**Figure 4 materials-14-00738-f004:**
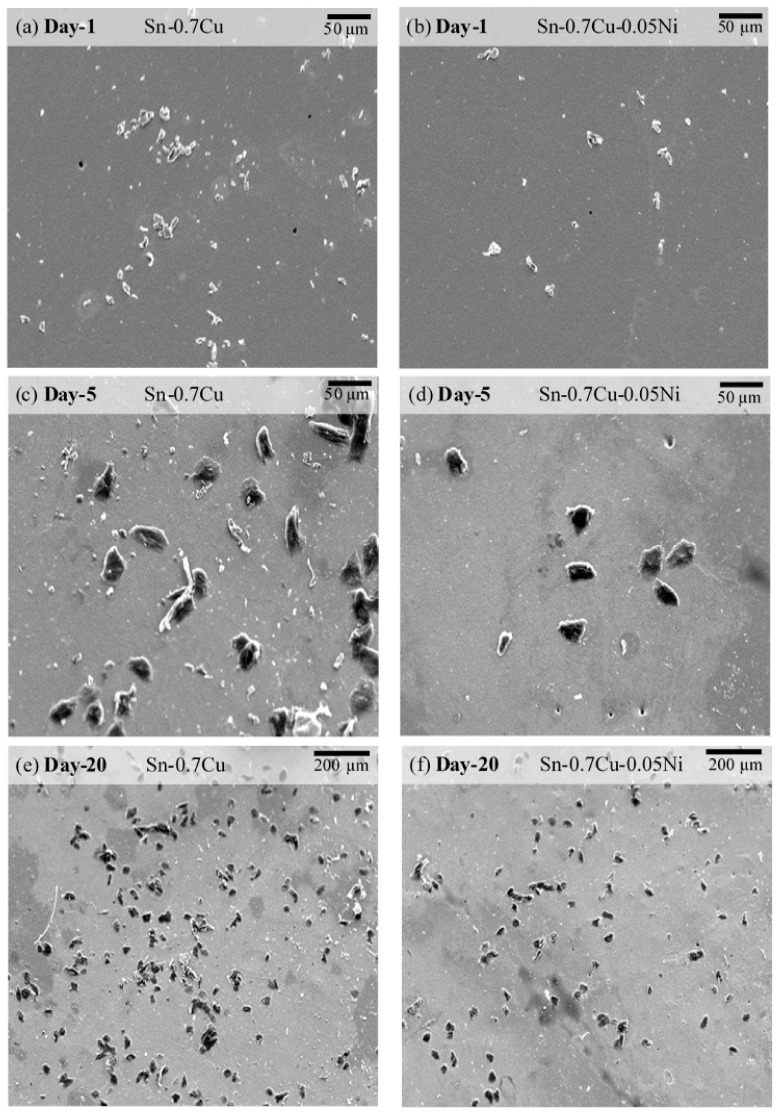
Surface morphologies of Sn whiskers’ distribution from Sn-0.7Cu and Sn-0.7Cu-0.05Ni solder coatings over a period of (**a**,**d**) 1 day, (**b**,**e**) 5 days and (**c**,**f**) 20 days.

**Figure 5 materials-14-00738-f005:**
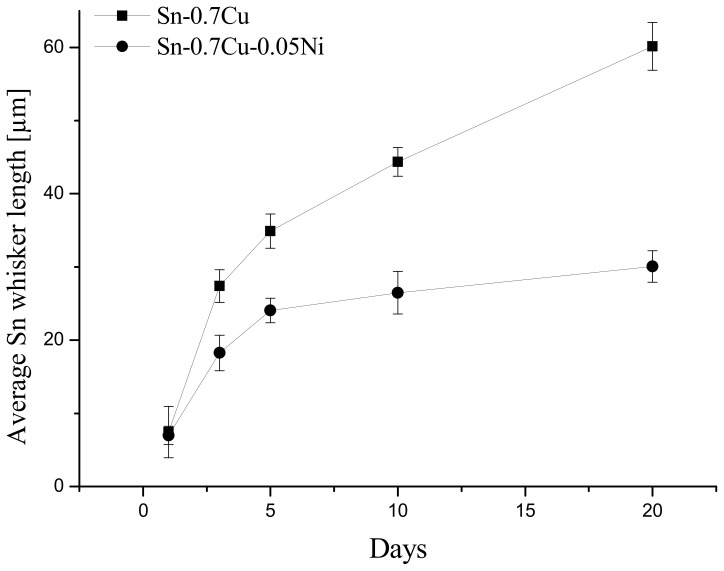
The average length of Sn whiskers in Sn-0.7Cu and Sn-0.7Cu-0.05Ni solder joints.

**Figure 6 materials-14-00738-f006:**
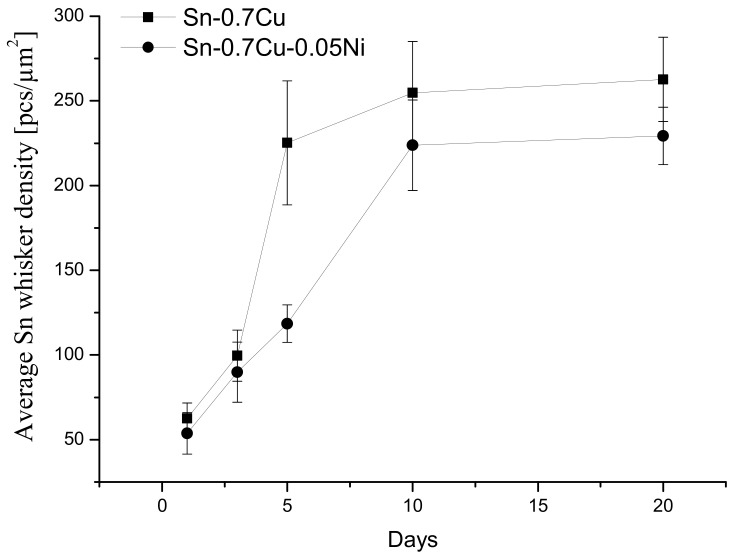
The average density of Sn whiskers in Sn-0.7Cu and Sn-0.7Cu-0.05Ni solder joints.

**Figure 7 materials-14-00738-f007:**
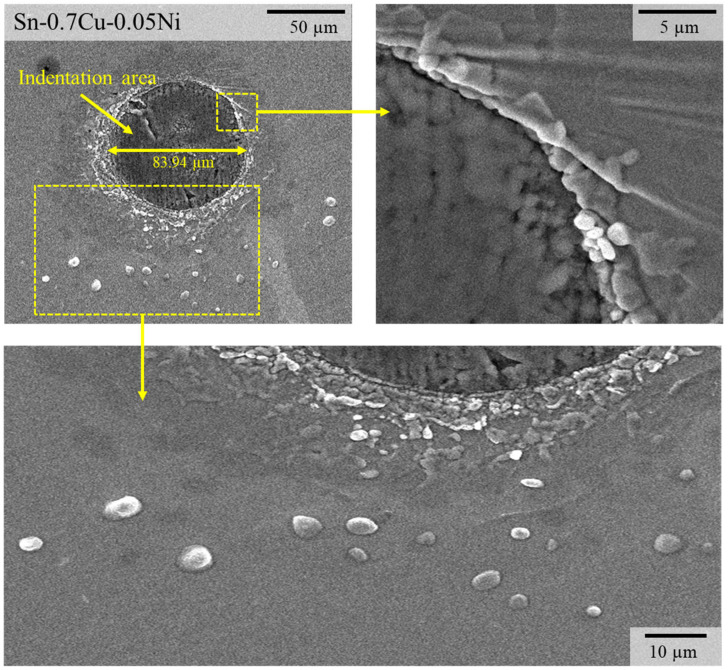
Scanning electron microscope (SEM) images of Sn whisker nucleation growth in the Sn-0.7Cu-0.05Ni solder joint after indentation for 12 h.

**Figure 8 materials-14-00738-f008:**
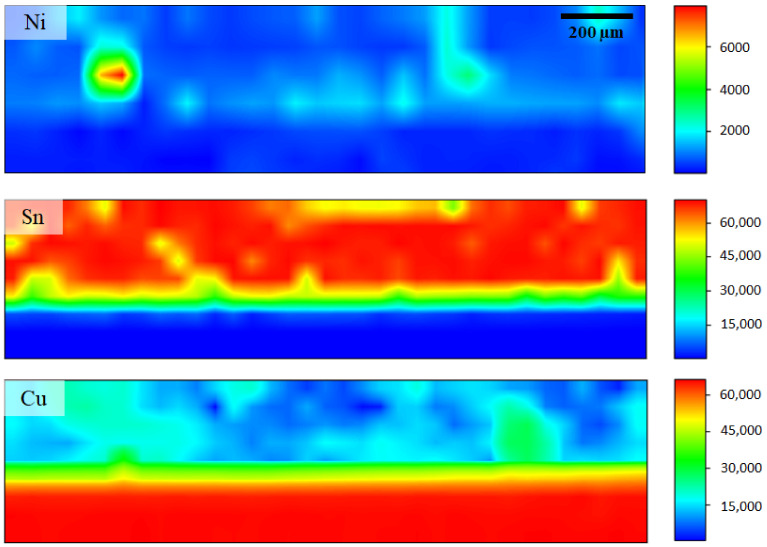
Elemental distribution of Sn-0.7Cu-0.05Ni on Cu substrate.

**Figure 9 materials-14-00738-f009:**
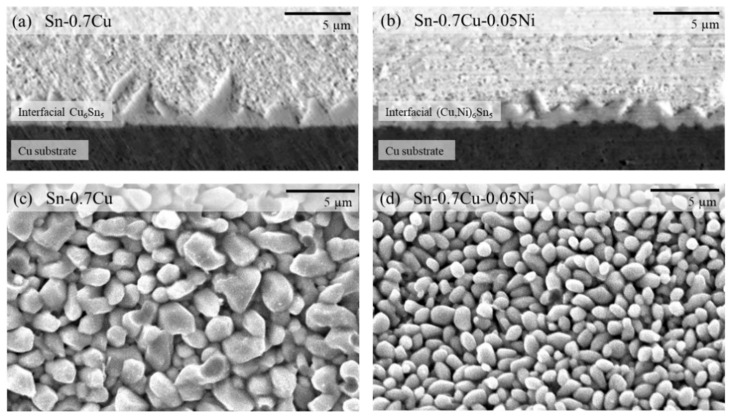
The morphology of the intermetallic compound (IMC) interfacial layer of Sn-0.7Cu and Sn-0.7Cu-0.05Ni solder joints. (**a**) Cross-section view of the interfacial Cu_6_Sn_5_, (**b**) cross-section view of the interfacial (Cu,Ni)_6_Sn_5_, (**c**) top view of the interfacial Cu_6_Sn_5_ and (**d**) top view of the interfacial (Cu,Ni)_6_Sn_5_.

**Figure 10 materials-14-00738-f010:**
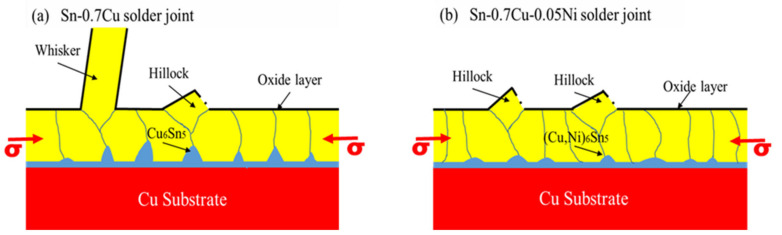
A schematic diagram illustrating the correlation between IMC interfacial growth and Sn whisker growth on (**a**) Sn-0.7Cu and (**b**) Sn-0.7Cu-0.05Ni.

**Figure 11 materials-14-00738-f011:**
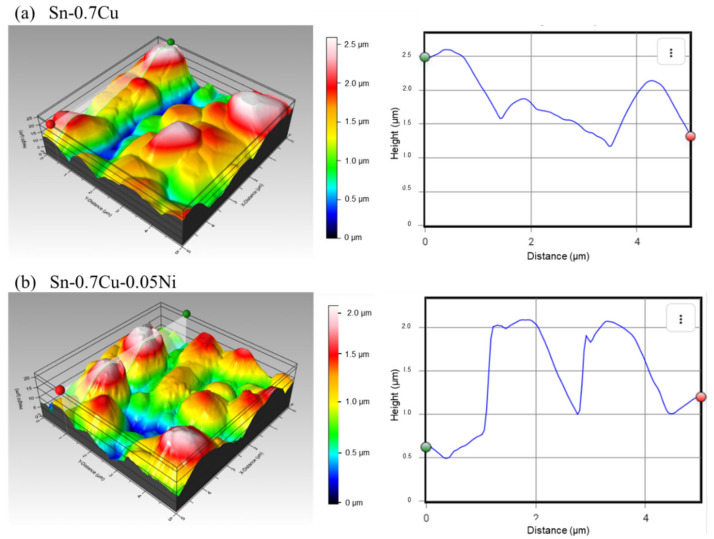
Atomic force microscope (AFM) three-dimensional (3D) images (5 × 5 µm) of interfacial IMC formed from (**a**) Sn-0.7Cu and (**b**) Sn-0.7Cu-0.05Ni.

**Table 1 materials-14-00738-t001:** The chemical composition analysis of lead (Pb)-free solder alloy and substrate (wt.%).

Composition	Sn-0.7Cu (wt.%)	Sn-0.7Cu-0.05Ni (wt.%)	Cu Substrate (wt.%)
Sn	99.2276	99.2184	–
Cu	0.7683	0.6563	99.9426
Ni	–	0.0528	–
Sb	0.0024	0.0064	–
Pb	0.0013	0.0162	0.0053
Zn	<0.0002	<0.0002	0.0142
Fe	0.0042	0.0041	–
Al	0.0003	0.0004	0.0148
In	0.0012	0.0034	–

**Table 2 materials-14-00738-t002:** Solder hot-dipping process parameters.

Hot-Dip Soldering Parameters	Value
Preheat Temperature (°C)	200
Preheat Time (s)	60
Preheat Rate (°C/s)	2.5
Peak Temperature (°C)	265
Immersion Withdrawal Speed (mm/s)	10
Immersion Dwell Time (s)	2

## Data Availability

The data presented in this study are available in this article.

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
