# Peer review of "Effect of Ni on the Suppression of Sn Whisker Formation in Sn-0.7Cu Solder Joint"

_materials, 2021, doi:10.3390/ma14040738_

Round 1

Reviewer 1 Report

This paper mainly studied the effect of Ni addition in Sn-Cu solder on Sn whisker formation.
It can be considered with the following considerations:

Table 1: Please insert the unit. (wt% or at%)

Fig. 7 : Please insert the images of Sn-0.7Cu solder joint case.

Fig. 9 : Please insert the intercacial images after 20 days storage.

Author Response

  1. Thank you for your comment. We have corrected the following comments. The unit of wt. % has been added into Table 1. - Page 3/line 116.
  2. Thanks for your constructive comments and opinion. Fig. 7 is emphasis on the effect of mechanical compressive stress induced of Sn whiskers growth in Sn-0.7Cu-0.05Ni solder joint that was indented for 12 hours.  

    For your information, there have no obvious effect on the distribution of Sn whisker growth on Sn-0.7Cu and Sn-0.7Cu-0.05Ni solder joint after indentation for 12 hours. Based on the imaged structures, hillock or nodule-shape Sn whiskers were observed on the both solder joint. Hillock growth appears to occur initially as nucleation followed by protrusion of Sn whiskers from the hillocks.

    With an increase of storage time of 120 hours to 480 hours (refer Fig. 4c-4f), the hillock-shape Sn whisker rapidly grew and become larger on the surface of the solder joint. In contrast, Sn whiskers growth were found greater on the Sn-0.7Cu solder with the growth of filament-shape and bent-shape Sn whisker, whereas the hillock-shape was observed on the surface of Sn-0.7Cu-0.05Ni solder joint.

    Figure 7 is mainly to show the example of indentation and area of whiskers growth on the sample. We believe that Sn-0.7Cu-0.05Ni solder joint can be a representative of the image. - Page 9/line 233-252
  3. Thank you for your constructive comments and opinion. Fig. 9 is focus on the effect of Ni addition on IMC interfacial growth of Sn–0.7Cu and Sn–0.7Cu-0.05Ni solder joint.  For your information, these images were observed without mechanical stress induced after 48 hours to saturated the residual internal stress in IMC interfacial layer.

    In addition, due to the limitation of scanning electron microscope (SEM) and the morphology of IMC interfacial layer that relatively thin (1.76 µm to 1.28 µm), it is difficult to measure the scientific progress of IMC interfacial growth rate with storage time. - Page 11/line 298-308

Reviewer 2 Report

Dear Authors,

I found your work interesting and well-developed and clearly exposed. The conclusions have scientific soundness and are supported by the data collected. The experimental design highlight also an interesting approach and the use of cutting edge technologies.

In my opinion, the paper needs minor spell corrections. Please, fix them.

Best Regards

Author Response

Thank you for your comment. We have corrected grammatical mistakes in this manuscript as mentioned by reviewers.  We also rewritten the text to improve the quality of manuscript. - All Pages

Reviewer 3 Report

Dear authors,

information about the new possibility to suppress the Sn whiskers growth, presented in your article, important and actual for design of the microelectronic circuits with high density of the elements. Newertheless, I'd like to give an advice for the further development of your research.

Development of advanced technology takes  huge efforts and time, to save  the time and the technological attempts it will be reasonable to use the  Molecular dynamics modeling method - similar to one, presented in:

Beilstein J. Nanotechnol.2020,11, 1776–1788.https://doi.org/10.3762/bjnano.11.160.

One more comment: in introduction  it would be reasonable to extend citation of the IMC - formation mechanism by some other, for example:

  • the formation of continuous layers of a liquid phase between solid grains:

J Mater Sci (2010) 45:2057–2061  ,  and references herein.

Author Response

  1. Thanks for your constructive comments and suggestions. We will consider this publication in our further research.

Reviewer 4 Report

Please check english grammar. Some examples of necessary english changes are:

Line 95: There is no reference for the "a axis".

Line 79: Change "has been" to "have been"

Line 92: Change "Ni able to decreases" to "Ni is able to decrease"

Line 127: Change "by immersed the substrate" to "by an immersion of the substrate"

Line 183: Change "a AFM data is able to generate...contains" to "AFM data can be used to generate...containing" 

Lines 190/191: Change "the emitted" to "that is emitted"

Line 199: Change "were" to "was"   

Line 205: Change "was resulted" to "resulted"

Line 206: Change "were" to "was"     

Line 218: I think the authors are not saying what they want to say. Perhaps, "their size increasing rate was slower as days went on"   

Line 258: Change "that the" to "the"

etc           

Author Response

  1. Thank you for your comment. We have corrected grammatical mistakes in this manuscript as mentioned.  We also rewritten the text to improve the quality of manuscipt.
